# Changes in Whole-Blood microRNA Profiles during the Onset and Treatment Process of Cerebral Infarction: A Human Study

**DOI:** 10.3390/ijms21093107

**Published:** 2020-04-28

**Authors:** Arata Abe, Mayui Tanaka, Akihito Yasuoka, Yoshikazu Saito, Shinji Okada, Masahiro Mishina, Keiko Abe, Kazumi Kimura, Tomiko Asakura

**Affiliations:** 1Department of Neurological Science, Graduate School of Medicine, Nippon Medical School, Tokyo 113-8603, Japan; 2Department of Applied Biological Chemistry, Graduate School of Agricultural and Life Sciences, The University of Tokyo, Tokyo 113-8657, Japan; 3Department of Neuro-pathophysiological imaging, Medicine, Nippon Medical School, Kawasaki 211-8533, Japan

**Keywords:** cerebral infarction, microRNA, hypertension, apoptosis

## Abstract

Circulating miRNA species are promising symptom markers for various diseases, including cardiovascular disease. However, studies regarding their role in the treatment process are limited, especially concerning cerebral infarction. This study aimed to extract miRNA markers to investigate whether they reflect both onset and treatment process of cerebral infarction. A total of 22 patients (P-group) and 22 control subjects (C-group) were examined for their whole-blood miRNA profiles using DNA GeneChip™ miRNA 4.0 Array, with six patients examined after treatment (T-group). A total of 64 miRNAs were found to be differentially expressed between the C- and P-groups. Out of 64 miRNAs, the expression levels of two miRNAs correlated with hypertension. A total of 155 miRNAs were differentially expressed between the P- and T-groups. Five common miRNAs were found among the 64 and 155 miRNAs identified. Importantly, these common miRNAs were inversely regulated in each comparison (e.g., C < P > T), including miR-505-5p, which was previously reported to be upregulated in aortic stenosis patients. Our previous study using rat cerebral infarction models detected the downregulation of an apoptosis repressor, WDR26, which was repressed by one of the five miRNAs. Our results provide novel information regarding the miRNA-based diagnosis of cerebral infarction in humans. In particular, the five common miRNAs could be useful makers for the onset and the treatment process. Trial registration: This study was registered in the UMIN Clinical Trials Registry (UMIN000038321).

## 1. Background

Cerebral infarction has the second-highest mortality among diseases and causes sequelae such as memory disorder and body paralysis in survivors [1]. Cerebral infarction can be classified into several types based on their cause, including atherosclerosis of large cerebral arteries (atherothrombotic), occlusion of perforator arteries (lacunar), and infarction caused by blood clots from other tissues (emboli). There are multiple risk factors reported, such as hypertension, diabetes mellitus, heart disease, dyslipidemia, and smoking [2]. Thus, the causal factors of cerebral infarction are closely linked with systemic health, which may be reflected by the circulating blood. One of the blood markers of cerebral infarction is S100 calcium-binding protein B (S100B), which is derived from astrocytes in the brain tissue [3,4]. S100B is reported to increase in response to other brain diseases [5]. Interleukin-6 and C-reactive protein are other candidates, however, their upregulation is also observed during general tissue inflammation [6,7]. As such, it is necessary to identify molecular markers with higher specificity for cerebral infarction.

MicroRNA (miRNA) species are noncoding RNAs approximately 20 base pairs in length found in body fluids. Within cells, miRNAs play a role in regulating gene expression via RNA-silencing mechanisms [8]. Some miRNAs exhibit changes in blood expression levels in response to various diseases. As such, they are expected to be useful diagnostic markers. Besides the significant number of reports on their use for the diagnosis of cancer and metabolic syndromes [9], relatively limited reports discuss their role in cerebral infarction [10,11,12,13], with these studies mainly focused on the morbidity of the patients, severe vs. mild symptoms, or patients vs. healthy subjects and no reports on the role of miRNAs in the treatment process, which may be equally valuable for medical use. In this study, we analyzed the blood miRNA profiles of cerebral infarction patients, focusing on the treatment process to obtain novel information regarding the molecular markers of cerebral infarction symptoms.

## 2. Results

### 2.1. Characteristics of the Subjects and the Study Procedure

Table 1 shows the symptoms and characteristics of the subjects. The blood samples were collected from seven subjects in the patient (P)-group, which consisted of 23 subjects, after treatment to form the treatment (T)-group. The 22 subjects that exhibited similar combinations of risk factors but no symptoms of cerebral infarction served as the control (C)-group. The detected miRNA signals were normalized in combinations of the C- vs. P-groups, the P- vs. T-groups, and the C- vs. T-groups (Figure 1) and subjected to principal component analysis (Appendix A). There was no significant segregation between the C- and P-groups in this comparison, while the P8 sample was solely plotted alongside both the PC1 and PC2 axes (Appendix A). The P- and T-groups showed relatively clear segregation, except for P13, who was the only subject treated with tissue plasminogen activator (Table 1 and Appendix A). As such, P8, P13, and T13 were omitted from subsequent analyses (Figure 1). The plots without P8, P13, and T13 were basically identical to the plots which included these data (Figure 2a–c). The C- and T-groups showed intermediate segregation between the C- vs. P-group and P- vs. T-group plots (Figure 2c).

### 2.2. Identification of Differentially Expressed miRNAs

To extract miRNAs that responded to the onset and the treatment process of cerebral infarction, we compared their expression levels between the C- and P-groups (*n* = 22 for each) using Welch’s test. For the comparison between the P- and T-groups (*n* = 6 for each), we used the paired t-test (Figure 1 and Figure 4) because the P- and T-groups were linked individually, enabling us to test the difference more strictly. As a result, 64 miRNAs (C vs. P) and 155 miRNAs (P vs. T) were detected in each test. In the C- vs. T-group comparison, the number of T > C miRNAs (129 genes) was approximately five times higher than that of T < C miRNAs (26 genes). The miRNA gene names are listed in Table 2 and Table 3, respectively. The expression levels of 64 miRNAs were subjected to hierarchical clustering analysis and represented in a heat map with dendrograms (Appendix A). We also aimed to find the correlation between the expression levels of differentially expressed miRNAs and the risk factors in the C- and P-groups. There were 11 subjects with hypertension in the C-group and 15 subjects in the P-group, while the numbers for the other risk factors were relatively small or unevenly distributed, making it impossible to obtain reliable data (Table 1). We identified two miRNAs, miR-4717-5p and miR-200a-5p, that showed significant lower expression levels in subjects with hypertension (*t*-test, *p* < 0.01) in the P-group (Table 2 and Figure 3).

### 2.3. Comparison of miRNA Profiles between Onset and the Treatment Process

We searched for overlapping miRNAs between the 155 miRNAs (P-group vs. T-group) and 64 miRNAs (C-group vs. P-group) and identified five miRNAs, namely, miR-505-5p, miR-1255b-5p, miR-550b-2-5p, miR-4523, and miR-6795-3p (the gray zones in Figure 4). Interestingly, their expressions were inversely regulated in each comparison, i.e., “C < P and P > T” or “C > P and P < T”. This indicated that the fluctuation patterns of these fives miRNAs corresponded to both the onset and the treatment process of cerebral infarction (Figure 5).

## 3. Discussion

We identified miRNA species that may reflect both the onset and the treatment process of cerebral infarction through comparisons between three groups, namely, subjects with no symptoms, patients, and treated patients. Among the detected miRNAs, four miRNA genes were previously reported to be correlated with cerebral infarction symptoms (Table 4). MiR-376a-3p was expressed at higher levels in the P-group than C-group, which was consistent with previous studies that reported its upregulation in cerebral infarction patients [14]. MiR-3184-5p showed lower expression levels in the P-group than the C-group and was also reported to be downregulated in cerebral infarction patients [12]. As such, it is highly possible that miR-376a-3p and miR-3184-5p could be used as markers for the onset of cerebral infarction symptoms. On the other hand, miR-941 exhibited lower expression levels in the T-group than in the P-group. Due to the fact that other research groups found miR-941 to be upregulated in cerebral infarction patients [15,16], this miRNA could be used as a specific marker for treatment. Although we identified miR-505-5p as a common miRNA in both the onset and the treatment process samples, there were some discrepancies regarding its expression pattern both in our and previous studies. Further human studies should be done to confirm the usefulness of these five common miRNAs as cerebral infarction markers.

Interestingly, we found that the expression levels of two miRNAs were significantly lower only in the P-group for patients with hypertension (Figure 3). Both were reported to be related to hypertension or cardiovascular disease [17,19,20,21,22,23,24], suggesting that these miRNAs could be used as markers for the development of cerebral infarction among patients with hypertension.

We detected a higher number of downregulated miRNAs (129 genes) than upregulated miRNAs (26 genes) associated with the treatment process (Table 3), indicating that a reduction in miRNA expression may be dominant in the treatment process. In addition, principal component analysis revealed weak segregation between the C- and T-groups (Figure 2), suggesting that the treatment did not revert the P-group to the C-group in terms of miRNA profile. Further experiments using other control data (e.g., subjects with no risk factors) should be done to identify treatment-marker miRNA species.

The roles of the five miRNAs common in both the onset and treatment processes were predicted by using the miRDB target scan web resource [25]. We previously identified blood mRNAs whose expression levels were affected in the rat infarction model [26]. Among these mRNAs, a monoamine transporter gene, Slc18a2, and the apoptosis repressor gene Wdr26 were included in the candidate target list (Table 5). Slc18a2 was upregulated in the rat infarction model and its candidate regulator, miR-1255b-5p, was also found to be upregulated in this study. Wdr26 was shown to be downregulated in the rat infarction model, while its candidate regulator, miR-550b-2-5p, was upregulated in this study, consistent with the inhibition model of miRNAs. Due to the fact that Wdr26 regulates the mitogen-activated kinase pathway to repress oxidative stress-induced apoptosis [27], it is possible that miR-550b-2-5p inhibits apoptosis in humans during cerebral infarction. However, it remains uncertain whether the five miRNAs are the cause or the result of cerebral infarction. Our future task is to measure inflammation markers and reactive oxygen species [28] in the blood of patients, which may provide some information regarding the relationship between these five miRNAs and the onset or treatment process of cerebral infarction.

## 4. Methods

### 4.1. Study Subjects

Subjects were recruited at the Nippon Medical School Musashi Kosugi Hospital, from April to December 2017. The subjects were classified into the patient group with cerebral infarction symptoms (P-group) or the control group without symptoms (C-group). Some of the P-group subjects were selected for comparison between pre-treatment and post-treatment, and their blood samples were collected after the treatment to form the treated group (T-group). The subjects were characterized by risk factors (hypertension, dyslipidemia, diabetes mellitus, atrial fibrillation, ischemic heart disease, and smoking habits), the period between the onset of symptoms and blood collection, National Institutes of Health Stroke Scale (NIHSS) on day of admission, stroke type (lacunar or emboli), modified Ranking Score (mRS), and treatment type (tissue plasminogen activator, edaravone, heparin, or ozagrel sodium), as listed in Table 1. All procedures were conducted under the approval of the ethics committee of Nippon Medical School and the University of Tokyo. Written, informed consent was obtained from all patients or from their next-of-kin. The project identification code is 334-28-31,approval date was October 1st 2016. This project was approved by the ethics committee or institutional review board in Nippon Medical School Musashi Kosugi Hospital Committee.

### 4.2. Sample Collection and RNA Preparation

Venous blood samples (2 mL) were collected in Venoject^®^ II tubes containing EDTA (Terumo Corporation, Tokyo, Japan) 1–12 h after onset. Samples were mixed with 6 mL TRIzol LS Reagent (Thermo Fisher Scientific, Waltham, MA, USA) and 1 mL RNase-free water within 2 h after collection and stored at −80 °C until RNA preparation. We adopted this method to maintain data consistency because the same method was applied to the miRNA analysis of rat blood in our previous study [26]. Total RNA was extracted by chloroform and precipitated by isopropanol following the manufacturer’s protocol. No additional purification procedure was applied to avoid loss of miRNA species. The concentration and purity of each sample were analyzed using the NanoDrop^®^ ND-1000 UV-Vis Spectrophotometer (Thermo Fisher Scientific, Waltham, MA, USA), RNA 6000 Nano Assay (Agilent Technologies, Santa Clara, CA, USA) and the Small RNA Assay (Agilent Technologies, Santa Clara, CA) on an Agilent 2100 Bioanalyzer (Agilent Technologies, Santa Clara, CA, USA). The RNA integrity numbers (RIN) of the RNA samples were lager than 7.0, thereby meeting the quality standard of the manufacturer.

### 4.3. DNA Microarray Analyis 

A biotin-labeled RNA probe was synthesized from 900 ng of total RNA from each sample using FlashTag Biotin HSR RNA Labelling Kit (Applied Biosystems, Foster City, CA, USA). Each probe was hybridized with a GeneChip™ miRNA 4.0 Array (Applied Biosystems, Foster City, CA, USA) and analyzed according to the manufacturer’s protocol. The hybridization experiments were conducted in the combinations (batches) described in Section 4.4. In the case of the T-group, all samples were subjected to the hybridization experiment at the same time. The signals detected by the probes were stored as CEL files.

### 4.4. Data Processing

The signal was normalized by the RMA method using the Transcriptome Analysis Console (ver. 4.0, Thermo Fisher Scientific, Waltham, MA, USA). For statistical analysis and principal component analysis, the JMP^®^ program (ver. 14, SAS Institute Inc., Cary, NC, USA) was used. Welch’s test (*p* < 0.01) was used to detect miRNAs differentially expressed between the C- and P-groups after normalization without P8 data (Figure 1, left flow). A paired t-test (*p* < 0.005) was used to detect miRNAs differentially expressed between the P- and T-groups after the normalization without P13 and T13 data (Figure 1, right flow). Normalization of the array signal was performed considering batch differences (Batch option on the Transcriptome Analysis Console). The constituents of each batch were as follows: Batch170803 (C11, C16, C27, P1, P3, and P4), Batch 170830 (C9, C10, C14, P5, and P7), Batch170908 (C1, C5, P9, and P11), Batch171005 (C13, C22, C23, P6, P13, and P14), Batch171018(C2, C3, C4, C12, C18, P2, P10, and P12), Batch180403 (C6, C20, C24, C25, P16, P17, P18, and P20), Batch180410 (C7, C17, P24, P25, and P27), and Batch180413 (P22 and P23). All samples in the T-group were of the same batch. For principal component analysis, data were normalized with the Batch option in the combination indicated by each plot (Figure 2a,b). Same data were subjected to statistical tests to detect differentially expressed miRNAs among the experimental groups (Welch’s t-test for C- vs. P-groups, paired t-test for P- vs. T-groups; Figure 1). Student’s t-test (*p* < 0.0005) was used to examine the difference in miRNA expression levels between subjects with and without hypertension. In this case, miRNA expression levels were normalized within the C- and P-groups, respectively. The cluster dendrogram with heat-mapping was drawn by JMP (https://www.jmp.com/ja_jp/offers/statistical-analysis-software.html) from the data presented in Figure 2a.

## 5. Conclusions

We identified multiple miRNA species whose expression levels were altered with onset and treatment of cerebral infarction. Five miRNAs in particular were found to be regulated oppositely in these processes, suggesting their potential use for predicting symptoms of cerebral infarction.

## Figures and Tables

**Figure 1 ijms-21-03107-f001:**
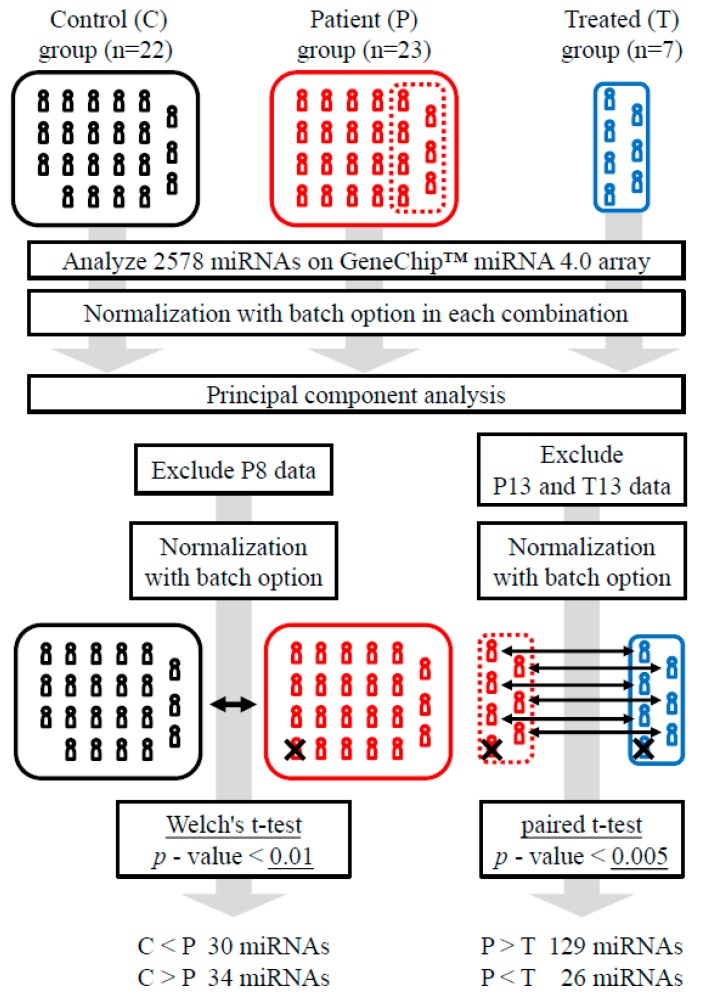
Flow chart of data processing. Normalization was conducted considering the batch differences (https://www.thermofisher.com/jp/en/home/life-science/microarray-analysis/microarray-analysis-instruments-software-services/microarray-analysis-software/affymetrix-transcriptome-analysis-console-software.html). See Section 4 for details.

**Figure 2 ijms-21-03107-f002:**
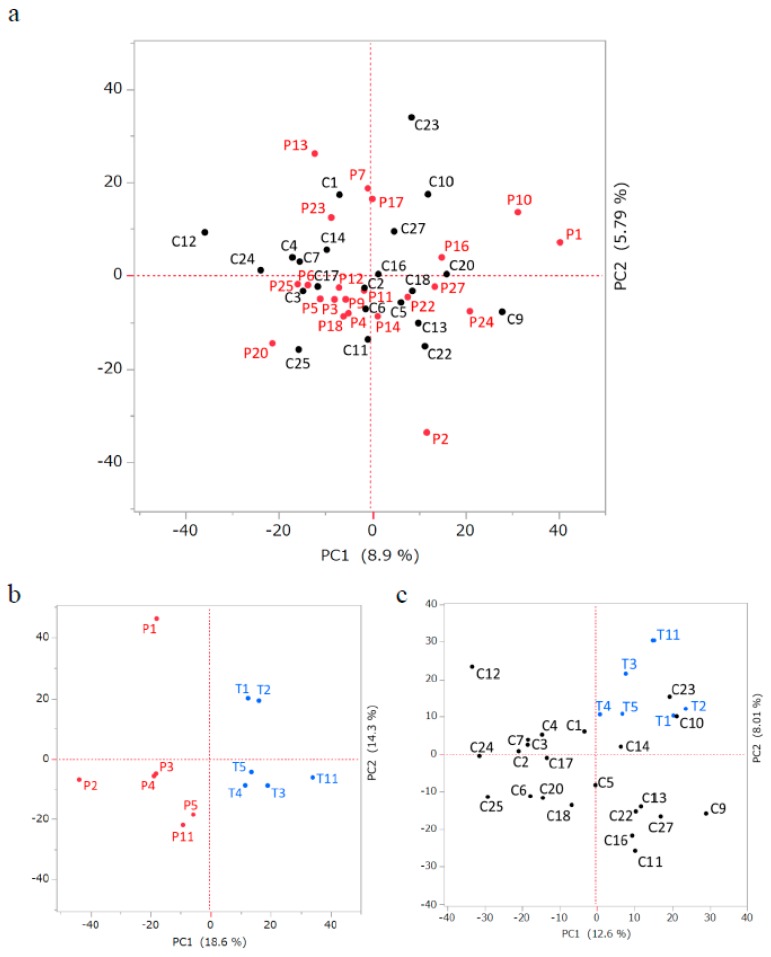
Principal component analysis of miRNA expression levels in each group comparison after exclusion of outlier data. The *X* and *Y* axes represent the first and the second principal components with contribution ratios in percentages. Control and patient groups (**a**), patient and treated group (**b**), and control and treated groups (**c**). “C”, “P”, or “T” with numbers indicate the samples in each group.

**Figure 3 ijms-21-03107-f003:**
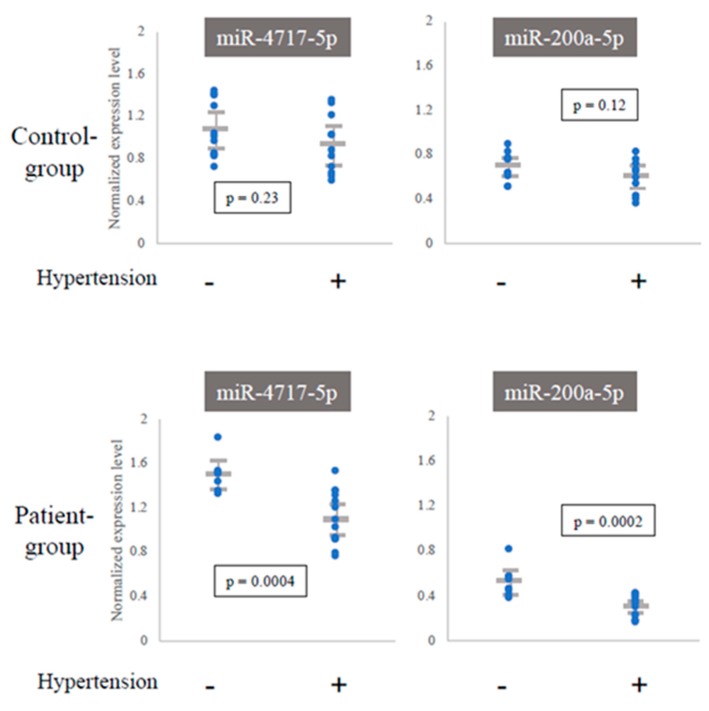
Dependence of miRNA expression levels on presence of hypertension. Expression levels of miRNAs were compared between subjects with hypertension and those without. The expression levels were normalized within each group. Significant differences (t-test, *p* < 0.01) were detected only in the P-group.

**Figure 4 ijms-21-03107-f004:**
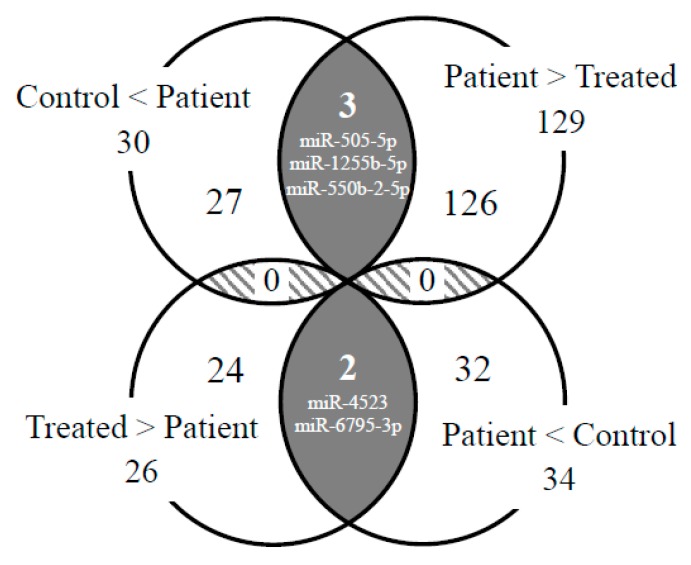
Venn diagram illustrating inclusion relationships of the detected miRNA sets. The numbers indicate the detected miRNAs. The numbers in the gray and hatched zones indicate miRNAs detected in both the control/patient and in the patient/treated comparisons.

**Figure 5 ijms-21-03107-f005:**
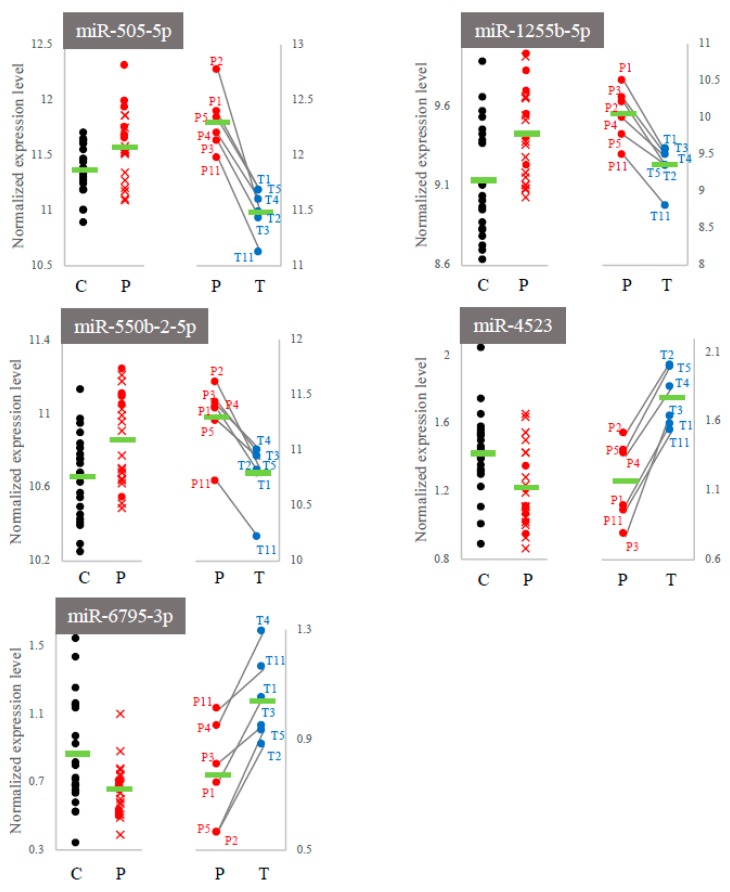
The expression levels of five common miRNAs within each experimental comparison. C and P: control group vs. patient group; P and T: patient group vs. treated group. The expression levels in the P-group were differentially represented due to differences in the normalization processes (Section 4).

**Table 1 ijms-21-03107-t001:** Background and symptoms of the subjects.

Sample	Age	Sex	Period between Occurrence and Blood Collection		Stroke Features				Treatment			Risk Factors					
			1st (P-group)	2nd (T-group)	Stroke Type		Severity	Post-Stroke Conditions	Tissue Plasminogen Activator	Edaravone	HeparinOzagrel-Sodium	Other Diseases					SM
					Lacunar	Emboli	NIHSS	mRS				HT	DM	DL	IHD	AF	
**P4 -> T4**	79	f	3	18		**+**	6	2			**+**	**+**	**+**	**+**			
**P3 -> T3**	74	m	3	21	**+**		4	2			**+**	+	**+**				
**P5 -> T5**	84	m	2	15		**+**	4	1		**+**	**+**	**+**	**+**				
**P11 -> T11**	78	m	1	18	**+**		1	0			**+**	**+**	**+**				
**P2 -> T2**	82	m	1	28		**+**	1	1		**+**	**+**		**+**		**+**		**+**
**P13 -> T13**	39	m	1	16	**+**		2	1	**+**	**+**	**+**			**+**			
**P1 -> T1**	73	m	1	13	**+**		5	1		**+**	**+**						**+**
P6	50	m	1		**+**		1					**+**	**+**	**+**			**+**
P8	63	f	1		**+**		1					**+**	**+**	**+**			**+**
P16	78	m	3		**+**		5					**+**	**+**	**+**	**+**	**+**	**+**
P27	70	m	2			**+**	12					**+**	**+**	**+**	**+**		
P20	50	f	2		**+**		3					**+**	**+**				
P7	65	m	5			**+**	7					**+**		**+**	**+**		**+**
P10	81	m	5		**+**		4					**+**		+		**+**	
P12	86	f	3			**+**	2					**+**		**+**			
P22	70	m	5		**+**		1					**+**		**+**			
P23	49	m	1		**+**		1					**+**		**+**			**+**
P9	77	f	1		**+**		5					**+**					**+**
P14	57	m	10		**+**		2						**+**				
P18	71	m	7			**+**	6						**+**	**+**	**+**		
P24	83	m	1			**+**	12						**+**			**+**	
P25	90	f	ND		**+**		6						**+**				
P17	28	m	2			**+**	7									**+**	
Total = 23	Ave.± SD = 68.6 ± 15.8	m/f =17/6			14	9	Ave.± SD = 4.26 ± 3.14	Ave.± SD = 1.14 ± 0.639	1	4	7	15	14	11	5	4	8
C16	75	m										**+**	**+**	**+**			**+**
C1	74	m										**+**	**+**		**+**		
C2	80	m										**+**	**+**				
C24	85	m										**+**	**+**				
C7	61	m										**+**		**+**			
C27	72	m										**+**			**+**		
C4	80	f										**+**					
C9	78	f										**+**					
C11	76	m										**+**					
C12	90	f										**+**					
C22	66	m										**+**					
C18	73	m											**+**				**+**
C6	51	m															**+**
C14	61	m															**+**
C3	79	m															
C5	83	m															
C10	82	m															
C13	40	m															
C17	26	m															
C20	55	f															
C23	46	m															
C25	89	f															
Total = 22	Ave.± SD = 69.2 ± 16.4	m/f = 17/5										11	5	2	2	0	4

HT: Hypertension / DL: Dyslipidemia / DM: Diabetes Mellitus / AF: Atrial fibrillation / IHD: Ischemic Heart Disease / SM: Smoking Habit.

**Table 2 ijms-21-03107-t002:** miRNAs identified by the comparison between the control and patient groups.

	C < P	C > P
miRNAs	let-7e-3p, miR-30d-3p, miR-302b-3p, miR-376a-3p, miR-505-5p, miR-514a-3p, miR-455-5p, miR-551b-3p, miR-411-3p, miR-1296-3p, miR-944, miR-548n, miR-1255b-5p, miR-3146, miR-3159, miR-3664-3p, miR-3689a-3p, miR-3908, miR-550b-2-5p, miR-4419b, miR-4518, miR-4662a-3p, miR-4717-5p, miR-4735-3p, miR-4764-3p, miR-5002-3p, miR-5582-5p, miR-6758-3p, miR-6810-5p, miR-6868-5p	miR-16-1-3p, miR-19a-5p, miR-27a-5p, miR-103a-3p, miR-181b-3p, miR-200a-5p, miR-516a-5p, miR-541-3p, miR-1233-3p, miR-1297, miR-1305, miR-1247-5p, miR-3156-5p, miR-3181, miR-3184-5p, miR-3194-5p, miR-4303, miR-3622b-3p, miR-3150b-5p, miR-3942-3p, miR-4502, miR-4523, miR-4665-3p, miR-4692, miR-4743-3p, miR-4772-5p, miR-5009-3p, miR-6506-5p, miR-6755-5p, miR-6795-3p, miR-6807-5p, miR-6856-5p, miR-7162-5p, miR-8079
Sum	30	34

**Table 3 ijms-21-03107-t003:** miRNAs identified by the comparison between the patient and treated groups.

	P > T	P < T
miRNAs	let-7b-5p, miR-25-5p, miR-29b-2-5p, miR-132-3p, miR-134-5p, miR-185-3p, miR-193a-5p, miR-320a, miR-200c-3p, miR-30c-1-3p, miR-130b-3p, miR-363-5p, miR-378a-3p, miR-342-5p, miR-331-5p, miR-339-3p, miR-423-5p, miR-432-5p, miR-501-3p, miR-502-3p, miR-505-5p, miR-532-3p, miR-574-3p, miR-550a-5p, miR-616-3p, miR-652-3p, miR-550a-3-5p, miR-1224-5p, miR-320b, miR-1271-5p, miR-1301-3p, miR-769-3p, miR-766-5p, miR-744-5p, miR-877-5p, miR-937-5p, miR-941, miR-942-3p, miR-1180-3p, miR-1226-3p, miR-1285-3p, miR-1287-5p, miR-1299, miR-1304-5p, miR-1254, miR-1270, miR-1275, miR-1292-5pe1255b-5p, miR-664a-5p, miR-1306-3p, miR-1307-3p, miR-2110, miR-2276-3p, miR-2278, miR-3124-5p, miR-3127-5p, miR-3136-5p, miR-3158-5p, miR-3164, miR-3173-3p, miR-3184-3p, miR-3200-5p, miR-3605-5p, miR-3615, miR-3619-3p, miR-3620-3p, miR-3667-5p, miR-3680-3p, miR-3682-3p, miR-3691-5p, miR-3150b-3p, miR-3928-3p, miR-3936, miR-3939, miR-3940-3p, miR-3944-5p, miR-550b-2-5p, miR-4433-3p, miR-4435, miR-4440, miR-4507, miR-4521, miR-4660, miR-4659b-5p, miR-4672, miR-4676-5p, miR-4723-5p, miR-4732-5p, miR-4732-3p, miR-3064-5p, miR-4738-3p, miR-4747-5p, miR-5010-5p, miR-5187-5p, miR-548aq-3p, miR-664b-5p, miR-6127, miR-6511a-3p, miR-6514-5p, miR-6515-5p, miR-6716-5p, miR-6511b-3p, miR-6734-5p, miR-6741-5p, miR-6743-3p, miR-6747-3p, miR-6749-3p, miR-6750-5p, miR-6756-5p, miR-6764-5p, miR-6767-5p, miR-6770-5p, miR-6799-3p, miR-6804-5p, miR-6815-5p, miR-6824-5p, miR-6831-5p, miR-6849-5p, miR-6859-5p, miR-6861-5p, miR-6877-5p, miR-6884-5p, miR-6890-5p, miR-6894-5p, miR-7113-5p, miR-7706, miR-7854-3p, miR-7855-5p	miR-20a-3p, miR-26b-5p, miR-30a-3p, miR-224-5p, miR-608, miR-622, miR-654-3p, miR-1181, miR-548p, miR-3622a-5p, miR-3687, miR-3913-5p, miR-3945, miR-4514, miR-4519, miR-4523, miR-3976, miR-4659a-3p, miR-4675, miR-5582-3p, miR-5684, miR-6131, miR-6744-5p, miR-6766-5p, miR-6795-3p, miR-6829-3p
Sum	129	26

The miRNAs underlined were also detected in the comparison between Control and Patient-groups.

**Table 4 ijms-21-03107-t004:** The miRNAs reported in previous studies.

miRNA	Comparison between the C- and P-Groups	Comparison between the P- and T-Groups	Previous Reports Related to Stroke
miR-376a-3p	C < P	Not Significant	Upregulated in ischemic stroke [14]
miR-3184-5p	C > P	Not Significant	Downregulated in ischemic stroke [12]
miR-941	Not Significant	P > T	Upregulated in patients with poor recovery from stroke compared to those with good recovery [15] Upregulated in potential ischemic stroke patients [16]
miR-505-5p	C < P	P > T	Downregulated in ischemic stroke [17] Upregulated in degenerative aortic stenosis [18]

“C < P”: Greater expression in the control group than in the patient group, as an example.

**Table 5 ijms-21-03107-t005:** Target gene candidates of the miRNAs detected in this study.

miRNA	Target Candidate	Predicted Regulation
miR-505-5p C < P and P > T	MR1, GLIS2, LAMP1, MECP2, CREBL2, CAPN5, ERCC1, ACAD11, SLAMF7, APOBEC3B, GDI2, ST3GAL1, AKT1S1, THUMPD2, CDK5, ATL2, PXT1, TRPM3, C5orf46, PSMD11, NSG2, VPS36, SIGLEC5, RARG, GNAO1, RETREG3, SDC3, SENP1, NEPRO, CREG1, HOXA3, NT5C2, MMP24, ZDHHC22, RAB7B, CLEC2A, TNIP1, TGM7, ATP8A2, BANF1, MDFIC, SLC8A3, SFTPB, SAPCD1, SMAGP, CAPSL, ATP1B2, SLC39A10, DLG2, PITPNM3, HBP1, GPR26, SYT15, POPDC2, SLC2A5, SMARCC1, ST8SIA3	C > P and P < T
miR-1255b-5p C < P and P > T	GSG1L, IREB2, FAM169B, C1orf185, DTX4, SUPT7L, ZNF420, STMN3, DDN, CXCL12, DIP2B, PDE6B, AVIL, PHYKPL, SERPINA11, DHRS7B, SLC18A2, ADAMTS3, PRDM10, PUS7L, TCHHL1, NEK11, SETBP1, JMJD8, ERLEC1, SCG3, RUNX1, SH3TC2, UBE2H, TBC1D5, MORN3, EPHA4, MORF4L1, FAM102A, GLYAT, NMNAT2, SEC63, FAM168A, HNMT, MFSD14B, ACSM6, CTNND2, BACH2, PPP2R1B, AKIRIN1, WDFY3, GPRC5A, YTHDF1, GNAL, TAF12, MBNL3, FNDC5, TRIM34, INSR, STRBP, PANK3, TCF24, ZDHHC22, CAMK2G, TRIM6-TRIM34, OSMR, YWHAZ, CAMTA1, CLEC4G, OAS2, MAP3K9, SDHAF3, NEU3, MID1, LANCL2, PCBP2, EPB41L1, CLEC4M, TIGAR	C > P and P < T
miR-550b-2-5p C < P and P > T	MAP1LC3B, KIAA1217, CDCA7, ANKRD13B, LRP1B, DSCAML1, ULK2, PHOX2B, CADM2, ADAM10, XKR6, CACNA1B, TSTD2, ATF2, TBL1XR1, GSK3B, SEPT8, KBTBD8, RSRC1, NXPH1, CPEB3, PBX1, USP3, FAM180B, CPNE4, NRXN1, MYH2, ADCY1, USP1, CYP27B1, B3GALT2, NDUFAF4, WDR26, YIPF3, RFESD, KLF6, SAMD12, PTPN21, ID4, FAM120A, NCBP1, SCN11A, ACTR3, HINT3, STXBP5, TMEM30B, CX3CR1, WAPL, EIF2S3, IL6ST	C > P and P < T
miR-4523 C > P and P < T	CSNK1A1L, FBXO8, GTF2F2, ABHD17C	C < P and P > T
miR-6795-3p C > P and P < T	ACSL6, IPO7, CXXC4, FAM241B, SPTBN4, TMEM189-UBE2V1, SOX9, UBE2V1, CNGB3, TET3, POLDIP3, B4GALT2, AOC3, SPIN1, MAST3	C < P and P > T

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
