# Peer review of "Changes in Whole-Blood microRNA Profiles during the Onset and Treatment Process of Cerebral Infarction: A Human Study"

_ijms, 2020, doi:10.3390/ijms21093107_

Round 1

Reviewer 1 Report

The study by Abe et al investigates the differential expression of regulatory miRNAs from peripheral blood following cerebral ischemia in human. The cohort consisted of 23 patients (P group) and 23 controls (22 after removal of an outlier, C group). miRNA expression was also measured in peripheral blood cells in 7 (6 after removal of an outlier) out of the 23 patients after treatment (T group). The authors identified a number of differentially expressed miRNA between the groups: 64 miRNAs between the C and P groups, and 155 miRNA between the P and T groups. The authors also investigated miRNAs associated with hypertension. The authors speculate that some of the differentially expressed miRNAs in the study might be good biomarkers

               I have several comments:

  • The abstract states that there were 23 control subjects and 7 treated subjects. However, 22 controls and 6 treated subjects were only analyzed due to removing of outliers. The more accurate numbers to be included in the abstract are 22 C and 6T subjects.
  • Please discuss the rational for using EDTA and not PAXgene tubes for blood collection. The latter ones stabilize the RNA, so there is no ex vivo gene expression occurring.
  • Page 4: the sentence “ Each probe was hybridized with GeneChip miRNA 4.0 Array….” should read “Each sample was hybridized with GeneChip miRNA 4.0 Array…”
  • The comparison should be set up Patient (P) vs Control (C), not C vs P, since the denominator is usually the control, to which a change in the gene expression is compared to.
  • Comparison to other relevant studies from literature is needed.
  • I could not evaluate the supplementary material, since it was not available in the downloaded file.
  • It seems Figure 2 represents the Principal Components Analyses plots (PCAs) on all the studied miRNA. However, similar PCAs should also be drawn on the differentially expressed genes to evaluate the robustness of the gene expression differences. Even more so important to perform these PCAs or even Hierarchical Clustering or Prediction Algorithms on the differentially expressed genes only, since the authors suggest that they might be good biomarkers. This can begin to be evaluated in this study (though it is a very small one).
  • Related to the PCA of the P and T groups in Figure 2b – it is problematic that all 6 samples of the T group were run in one batch as stated in the Methods section. Group separation (as seen in Fig 2b) is not typical at a whole genome (in this case whole miRNA-ome) level. This separation observed on the PCA plot may be due to a batch effect, which a batch remover cannot remove because there is complete confounding of batch for the T group.
  • In Table 1, please represent the average time +/- SD from onset to blood draw for the P and T groups, since the immune response is very dynamic and it is important to know what time window the study was performed on.
  • Figure 3 is not referenced in the text and is out of order.
  • The overlapping miRNA in Figure 4 as the authors state have a fluctuating pattern corresponding to both response to the onset and the treatment. For ex. miR-5050-5p, miR-1225b-5p and miR-5506-2-5p have higher levels in the Patient group than in the Control group, and after treatment ( the T group) their levels decrease. It would be interesting to know whether their levels go back to “base”/control’s level. Similarly, for the other 2 miRNAs mentioned in the overlap – miR-4523 and miR-6795-3p, which have C>P<T pattern of expression. After treatment, do their levels go up to the Control’s level? Figure 5 visually addresses that, but due to the differences in the normalization process, as the authors state in the figure legend, the reader cannot conclude the answer to my question A direct comparison for these 5 miRNAs between the C and T groups can answer that.
  • Pathway analysis on the predicted targets might shed a light into the processes these miRNAs control.
  • English needs to be revised, since there are sentences/paragraphs where meaning is unclear.

Author Response

Comment 1

 The abstract states that there were 23 control subjects and 7 treated subjects. However, 22 controls and 6 Must be Not treated subjects were only analyzed due to removing of outliers. The more accurate numbers to be included in the abstract are 22 C and 6T subjects.

Answer 1

 Thank you for your indication. We corrected the abstract as follows.

" ABSTRACT:  Circulating miRNA species are becoming promising symptom markers for various diseases including cardiovascular diseases. However, studies on their role in the treatment processes are few especially concerning cerebral infarction. This study aimed to extract miRNA markers that may reflect both onset and treatment process of cerebral infarction. 22 patients (P-group) and 22 control subjects (C-group)were examined for their whole blood miRNA profiles using DNA GeneChip™ miRNA 4.0 Array. 6 of the patients were examined after the treatment (T-group). A total of 64 miRNAs were found to be differentially expressed between C and P-groups. Out of 64 miRNAs, the expression levels of 2 miRNAs correlated with hypertension. A total of 155 miRNAs were differentially expressed between P- and T-group. There were 5 common miRNAs among the 64 and 155 miRNAs identified. Importantly, these 5 common miRNAs were inversely regulated in each comparison (e.g., C < P > T). These included miR-505-5p, reported to be up-regulated in aortic stenosis patients. Our previous study using rat cerebral infarction models detected the downregulation of an apoptosis repressor, WDR26, which can be repressed by one of the 5 miRNAs. Our results provide novel information for the miRNA-based diagnosis of cerebral infarction in humans, especially, the 5 common miRNAs can be useful makers for the onset and the treatment process."

Comment 2

 Please discuss the rational for using EDTA and not PAXgene tubes for blood collection. The latter ones stabilize the RNA, so there is no ex vivo gene expression occurring.

Answer 2

 We corrected the sentences in Methods L77-81 as follows.

Original

"A sample of 2 mL of venous blood was collected from the subjects and was put into a vacuum blood collection tube Venoject® II containing EDTA (Terumo Corporation, Tokyo, Japan). Samples were gently mixed and subjected to the next step within 2 hrs. For RNA extraction, 1 mL of each blood sample was mixed well with 6mL TRIzol LS Reagent (Thermo Fisher Scientific, Waltham, MA, USA) and 1mL RNase-free water and stored at -80 °C until RNA preparation."

Corrected

" 2 mL of venous blood samples were collected in Venoject® II tubes containing EDTA (Terumo Corporation, Tokyo, Japan) 1 to 12 hours after the onset. Samples were mixed with 6mL TRIzol LS Reagent (Thermo Fisher Scientific, Waltham, MA, USA) and 1mL RNase-free water within 2 hours after the collection and stored at -80 °C until RNA preparation. We adopted this method to maintain the consistency of data because the same method has been applied to the miRNA analysis of rat blood in our previous study [23]."

Comment 3

 Page 4: the sentence “ Each probe was hybridized with GeneChip miRNA 4.0 Array....” should read “Each sample was hybridized with GeneChip miRNA 4.0 Array...”

Answer 3

 We corrected the sentences in Methods L90-93 as follows.

Original

"Biotin-labeled RNA probes were synthesized from 900 ng of total RNA using FlashTag Biotin HSR RNA Labelling Kit (Applied Biosystems, Foster City, CA). Each probe was hybridized with GeneChip™ miRNA 4.0 Array (Applied Biosystems, Foster City, CA) and analyzed according to the manufacturer’s protocol."

Corrected

"Biotin-labeled RNA probe was synthesized from 900 ng of total RNA from each sample using FlashTag Biotin HSR RNA Labelling Kit (Applied Biosystems, Foster City, CA). Each probe was hybridized with GeneChip™ miRNA 4.0 Array (Applied Biosystems, Foster City, CA) and analyzed according to the manufacturer’s protocol."

Comment 4

 The comparison should be set up Patient (P) vs Control (C), not C vs P, since the denominator is usually the control, to which a change in the gene expression is compared to.

Answer 4

 Thank you for your indication. We corrected the expression throughout the manuscript.

Comment 5

 Comparison to other relevant studies from literature is needed.

Answer 5

 We summarized the results of other relevant studies in Table 4 and discussed about them.

Comment 6

 I could not evaluate the supplementary material, since it was not available in the downloaded file.

Answer 6

 We sincerely apologize for the fault of our manuscript. Please see attached Supplementary Figure 1.

Comment 7

 It seems Figure 2 represents the Principal Components Analyses plots (PCAs) on all the studied miRNA. However, similar PCAs should also be drawn on the differentially expressed genes to evaluate the robustness of the gene expression differences. Even more so important to perform these PCAs or even Hierarchical Clustering or Prediction Algorithms on the differentially expressed genes only, since the authors suggest that they might be good biomarkers. This can begin to be evaluated in this study (though it is a very small one).

Answer 7

 According to your suggestion, we drew cluster dendrogram and heat map from DEG data (ResFig.1_P vs C Hierarchical clustering). The subjects were classified into 7 clusters at the 4th nodes (A to G in ResFig.1_P vs C Hierarchical clustering). However these clusters did not show correlation to any of parameters attributed to the individuals (ResTable1_Factors_Cluster). From these results, we conclude that this analysis is not useful as statistic tests that we adopted to identify specific miRNA such as hypertension-correlated miRNAs (Fig. 5) and overlapping miRNAs (Fig. 4).Your kind understanding will be greatly appreciated.

Comment 8

 Related to the PCA of the P and T groups in Figure 2b – it is problematic that all 6 samples of the T group were run in one batch as stated in the Methods section. Group separation (as seen in Fig 2b) is not typical at a whole genome (in this case whole miRNA-ome) level. This separation observed on the PCA plot may be due to a batch effect, which a batch remover cannot remove because there is complete confounding of batch for the T group.

Answer 8

 The Batch option described in Methods reduces deviation caused by batch differences and allows statistic comparison between groups composed with different batches. Accordingly, we conducted normalization with Batch option in every combination before PCA and statistic comparison of miRNA expression. To help understanding of readers, we added following description to Methods L111.

"For principal component analysis, data were normalized with Batch option in the combination indicated by each plot (Fig. 2 a to b). Same data were subjected to the statistic tests for the detection of differentially expressed miRNAs among the experimental groups (Welch's t-test for C vs. P, paired t-test for P vs. T, Figure 1)."

Comment 9

 In Table 1, please represent the average time +/- SD from onset to blood draw for the P and T groups, since the immune response is very dynamic and it is important to know what time window the study was performed on.

Answer 9

 Unfortunately, we have no individual record for sampling time. We corrected the description in Methods L77 as in Answer 2.

Corrected

" 2 mL of venous blood samples were collected in Venoject® II tubes containing EDTA (Terumo Corporation, Tokyo, Japan) 1 to 12 hours after the onset. Samples were mixed with 6mL TRIzol LS Reagent (Thermo Fisher Scientific, Waltham, MA, USA) and 1mL RNase-free water within 2 hours after the collection and stored at -80 °C until RNA preparation. We adopted this method to maintain the consistency of data because the same method has been applied to the miRNA analysis of rat blood in our previous study [23]."

Comment 10

 Figure 3 is not referenced in the text and is out of order.

Answer 10

 We have referred Figure 3 in Results 3.2. Identification of differentially expressed miRNA L153 and Discussion L197.

Comment 11

 The overlapping miRNA in Figure 4 as the authors state have a fluctuating pattern corresponding to both response to the onset and the treatment. For ex. miR- 5050-5p, miR-1225b-5p and miR-5506-2-5p have higher levels in the Patient group than in the Control group, and after treatment ( the T group) their levels decrease. It would be interesting to know whether their levels go back to “base”/control’s level. Similarly, for the other 2 miRNAs mentioned in the overlap – miR-4523 and miR- 6795-3p, which have C>P<T pattern of expression. After treatment, do their levels go up to the Control’s level? Figure 5 visually addresses that, but due to the differences in the normalization process, as the authors state in the figure legend, the reader cannot conclude the answer to my question A direct comparison for these 5 miRNAs between the C and T groups can answer that.

Answer 11

 The values used in Figure 5 have been normalized in each combination of experimental groups (C vs. P or P vs. T). As you suggested, it is possible to obtain normalized values in C vs. T combination. However, the number of samples are not equal among C and T-group (22 and 6, respectively), which makes it difficult to perform statistic test. We can say at least that C and T-groups are not equal in terms of overall expression profile as indicated by their PCA in Figure 2c. Your kind understanding will be greatly appreciated.

Comment 12

 Pathway analysis on the predicted targets might shed a light into the processes these miRNAs control.

Answer 12

 Thank you for your evaluation. We are going to confirm the significance of these target molecules.

Comment 13

 English needs to be revised, since there are sentences/paragraphs where meaning is unclear.

Answer 13

 Thank you for your indication. We have improved unclear expressions according to you suggestion.

Reviewer 2 Report

The study by Abe et al., attempts to determine microRNA markers in human cerebral infarction patients. As the authors correctly state, cerebral infraction is amongst the leading causes of death and disability worldwide. Due to the stability of microRNAs in blood they are considered as good candidate markers for understanding IS pathology and treatment within the limited time window for intervention that the IS allows for.

In this study, the authors collected whole blood miRNA profiles from IS patients before and after IS treatment. Although the rationale of the study is solid, many factors including the presentation and design of the study, in addition to the attention the authors give to spell and grammatical error-checking, impacts the overall image of the study.

1. Specifically, the authors have not done spell checking and grammatical correction of their abstract and main paper.

For example:

Line 18: "This study aimed to extract miRNA 18 markers that may reflect both onset and teratment process of cerebral infarction."

Line 21: "A Total of 64 miRNAs were found..."

Line 21/22: "Of 64 miRNAs,.." should be "Out of..."

Line 169: "...previously identified and identified 5 miRNAs, miR-505-5p, miR-1255b-5p,..."

2. Although the approach the authors adopt in evaluating their hypothesis there are a few shortcomings in the experimental carryout and presentation that affect the whole image of the study. For example, the recruited subjects belong to a wide range of ages i.e., P 28-90, T26-90, nonetheless only one subject in its 20s is included. A skewed age range in subjects will impact the results as it has been shown that the expression profiles in young IS patients are different (Tan et al. PLOS ONE 2009). Along the same lines, in the treated group (T) only one female is included and the number of treated subjects is quite low compared to the other groups.

3. Another contradicting point of this study is that the process by which the analysis of the results was done is not clearly presented and/or stated making it hard to follow the flow of the text/results.

Author Response

Comment 1

 Line 18: "This study aimed to extract miRNA 18 markers that may reflect both onset and teratment process of cerebral infarction."

Line 21: "A Total of 64 miRNAs were found..."
Line 21/22: "Of 64 miRNAs,.." should be "Out of..."

Line 169: "...previously identified and identified 5 miRNAs, miR-505-5p, miR-1255b-5p,..."

Answer 1

 Thank your for your indication. We corrected the manuscript as follows.

" ABSTRACT:  Circulating miRNA species are becoming promising symptom markers for various diseases including cardiovascular diseases. However, studies on their role in the treatment processes are few especially concerning cerebral infarction. This study aimed to extract miRNA markers that may reflect both onset and treatment process of cerebral infarction. 22 patients (P-group) and 22 control subjects (C-group)were examined for their whole blood miRNA profiles using DNA GeneChip™ miRNA 4.0 Array. 6 of the patients were examined after the treatment (T-group). A total of 64 miRNAs were found to be differentially expressed between C and P-groups. Out of 64 miRNAs, the expression levels of 2 miRNAs correlated with hypertension. A total of 155 miRNAs were differentially expressed between P- and T-group. There were 5 common miRNAs among the 64 and 155 miRNAs identified. Importantly, these 5 common miRNAs were inversely regulated in each comparison (e.g., C < P > T). These included miR-505-5p, reported to be up-regulated in aortic stenosis patients. Our previous study using rat cerebral infarction models detected the downregulation of an apoptosis repressor, WDR26, which can be repressed by one of the 5 miRNAs. Our results provide novel information for the miRNA-based diagnosis of cerebral infarction in humans, especially, the 5 common miRNAs can be useful makers for the onset and the treatment process."

" We searched for overlapping miRNAs between the 155 miRNAs (P vs T) and 64 miRNAs (C vs P) previously identified and identified 5 miRNAs,"

Comment 2

 Although the approach the authors adopt in evaluating their hypothesis there are a few shortcomings in the experimental carryout and presentation that affect the whole image of the study. For example, the recruited subjects belong to a wide range of ages i.e., P 28-90, T26-90, nonetheless only one subject in its 20s is included. A skewed age range in subjects will impact the results as it has been shown that the expression profiles in young IS patients are different (Tan et al. PLOS ONE 2009). Along the same lines, in the treated group (T) only one female is included and the number of treated subjects is quite low compared to the other groups.

Answer 2

 We would like to conserve the data because the subject number in each group was relatively low. Basically, these low number of exceptional subjects may increase the sample deviation, resulting in detection of less number of DEGs. Even though this is the case, we were able to detect significant number of DEGs that showed correlation to the onset and treatment processes. Your kind understanding will be greatly appreciated.

Comment 3

 Another contradicting point of this study is that the process by which the analysis of the results was done is not clearly presented and/or stated making it hard to follow the flow of the text/results.

Comment 3

 The process of data selection and statistic test were illustrated by Figure 1. We hope it helps reader's understanding.

Round 2

Reviewer 1 Report

The authors have addressed most of my comments. However, there are a couple more that they have not addressed entirely:

  • The Hierarchical Clustering of the differentially expressed genes that the authors performed in response to my comments should be included as a figure in the paper, not just as a figure sent to me. Even thought the authors conclude in their response to my comment that “However these clusters did not show correlation to any of parameters attributed to the individuals (ResTable1_Factors_Cluster)”, they still do show separation of the P and C groups. The authors do not need to include their labeling of the clusters A to G, just show the Hierarchical Clustering dendrogram with the notation of the P and C subjects. This is a good result showing separation of the P and C groups and is one of the standard methods for visualizing differentially expressed gene signatures.
  • The authors must have misunderstood comment 12 in my original review. I was suggesting looking into the predicted targets of the differentially expressed miRNA and seeing what Canonical Pathways or Gene Ontology categories they are over-represented in.

Author Response

Answer to Reviewer 1

Comment 1

 The Hierarchical Clustering of the differentially expressed genes that the authors performed in response to my comments should be included as a figure in the paper, not just as a figure sent to me. Even thought the authors conclude in their response to my comment that “However these clusters did not show correlation to any of parameters attributed to the individuals (ResTable1_Factors_Cluster)”, they still do show separation of the P and C groups. The authors do not need to include their labeling of the clusters A to G, just show the Hierarchical Clustering dendrogram with the notation of the P and C subjects. This is a good result showing separation of the P and C groups and is one of the standard methods for visualizing differentially expressed gene signatures.

Answer 1

 Thank you for your valuable suggestion. We added a hierarchical clustering dendrogram as Supplementary Figure 2 and relevant description to Results and Methods.

Comment 2

 The authors must have misunderstood comment 12 in my original review. I was suggesting looking into the predicted targets of the differentially expressed miRNA and seeing what Canonical Pathways or Gene Ontology categories they are over-represented in.

Answer 2

 Thank you for your valuable suggestion. We analyzed the target candidate genes with their predicted change in expression (please see revised Table 5). However, we had a difficulty to explain any stroke related symptoms based on the predicted pathways (see attached files in IPA stroke target gene). Please let us omit these results from the manuscript.

Reviewer 2 Report

Comment 1

Although the authors corrected the grammatical and spelling errors in their first version of the manuscript the revised version still has grammatical errors

L77: A sample of 2 mL of venous blood wassamples were collected from

L88: into a vacuum blood collection tubein Venoject® II tubes containing

L79 Tokyo, Japan).) 1 to 12 hours after the onset

L80 each blood sample was mixed wellmixed with 6mL

L93 Biotin-labeled RNA probes wereprobe was synthesized

L174-175 We searched for overlapping miRNAs between the 155 miRNAs (P vs T) and 64 miRNAs (C vs P) previously identified and identified 5 miRNAs

Comment 2

i. With respect to the samples used and to the PCA analysis (figs 2b and 2c): by looking at the PCA analysis (fig 2) there is a substantial segregation of samples from the treatment (T1 and T2) and patient group (P1) in the respective PCA and of the control (C23, C10, C14) and treatment groups PCA. How do the authors treat this segregation considering the low number of samples? It seems that even within the groups there is a considerate amount of variability that makes the interpretation of the findings difficult without additional controls (see comment 3).

ii. with respect to the reply of the authors about the low number of samples:

"We would like to conserve the data because the subject number in each group was relatively low. Basically, these low number of exceptional subjects may increase the sample deviation, resulting in detection of less number of DEGs. Even though this is the case, we were able to detect significant number of DEGs that showed correlation to the onset and treatment processes".

I understand that finding subjects that fit into a clinical trial profile is difficult but if the low number of subjects resulted in high sample deviation (which could be also seen in the PCA analysis) then the DEG results would need additional controls to be believable and more solid. Specifically, the n-number in the treated group must be increased and see comment 3

Comment 3

The authors need to validate the observed DEGs by using another type of control such as quantitative PCR and also try to show that indeed some of the predicted targets shown in table 5 are indeed valid by again using either a qPCR and/or protein quantitation approach and perhaps an in vitro system.

Comment 4

What do the authors mean by discrepancies of the miRNA expression in their study?

L196-198 Although we identified miR-505-5p as a common miRNA between the onset and the treatment process, there were some discrepancies regarding its expression pattern both in our and previous studies.

How can then miR-505-5p be considered a reliable result?

Author Response

Answer to Reviewer 2

Comment 1

 Although the authors corrected the grammatical and spelling errors in their first version of the manuscript the revised version still has grammatical errors

L77: A sample of 2 mL of venous blood wassamples were collected from

L88: into a vacuum blood collection tubein Venoject® II tubes containing

L79 Tokyo, Japan).) 1 to 12 hours after the onset

L80 each blood sample was mixed wellmixed with 6mL

L93 Biotin-labeled RNA probes wereprobe was synthesized

Answer 1

 We sincerely apologize for mistakes in our manuscript. We have corrected all of them.

Comment 2

 With respect to the samples used and to the PCA analysis (figs 2b and 2c): by looking at the PCA analysis (fig 2) there is a substantial segregation of samples from the treatment (T1 and T2) and patient group (P1) in the respective PCA and of the control (C23, C10, C14) and treatment groups PCA. How do the authors treat this segregation considering the low number of samples? It seems that even within the groups there is a considerate amount of variability that makes the interpretation of the findings difficult without additional controls (see comment 3).

Comment 3

 with respect to the reply of the authors about the low number of samples:

"We would like to conserve the data because the subject number in each group was relatively low. Basically, these low number of exceptional subjects may increase the sample deviation, resulting in detection of less number of DEGs. Even though this is the case, we were able to detect significant number of DEGs that showed correlation to the onset and treatment processes".

I understand that finding subjects that fit into a clinical trial profile is difficult but if the low number of subjects resulted in high sample deviation (which could be also seen in the PCA analysis) then the DEG results would need additional controls to be believable and more solid. Specifically, the n-number in the treated group must be increased and see comment 3

Answer 2 and 3

 As you pointed out, subject number was not larger than those of other clinical studies and there were deviations in samples within P, C, and T-groups. However, the thresholds for our tests (p<0.01 for C vs P and p<0.005) may be more strict than many cases (p<0.05), supporting the solidity of our study. It is true that our result needs confirmation in clinical site with more number of subjects. At least, our study was able to provide 5 good candidate out of over 2000 miRNAs. Your kind understanding will be greatly appreciated.

Comment 4

 The authors need to validate the observed DEGs by using another type of control such as quantitative PCR and also try to show that indeed some of the predicted targets shown in table 5 are indeed valid by again using either a qPCR and/or protein quantitation approach and perhaps an in vitro system.

Answer 4

 It is a good idea to confirm expression levels of 5 miRNAs using qPCR. However, it is not possible to conduct experiment in our lab due to the delayed international transportation by COVID-19, and there is no good sign of recovery. We hope you understand that our findings add a valuable knowledge to this field.

Comment 5

 What do the authors mean by discrepancies of the miRNA expression in their study?

L196-198 Although we identified miR-505-5p as a common miRNA between the onset and the treatment process, there were some discrepancies regarding its expression pattern both in our and previous studies.

How can then miR-505-5p be considered a reliable result?

Answer 5

 As we described in Table 4, one of the studies reported the regulation of miR-505-5p opposite to our observation (Int J Mol Sci, 2014). We have no rationale for this phenomena, but all the other 5 reports showed no contradiction to our results. This overall ratio (1 vs 5) may support the reliability of our results.

Round 3

Reviewer 2 Report

Dear Authors,

due to the unprecedented Covid-19 situation and its effects on science, I understand the difficulty of conducting further experiments, thus I would not insist on point 4 but please do try to follow up and enriching this study with more samples and variation in samples and also in validation of the targets found.

Best,

Author Response

We thank for your kind understanding. We hope your research will be successful under this situation.